# Antimicrobial Stewardship at Transitions of Care to Outpatient Settings: Synopsis and Strategies

**DOI:** 10.3390/antibiotics11081027

**Published:** 2022-07-30

**Authors:** Elaine Liu, Kristin E. Linder, Joseph L. Kuti

**Affiliations:** 1Department of Pharmacy Services, Hartford Healthcare, Hartford, CT 06106, USA; elaine.liu121@gmail.com (E.L.); kristin.linder@hhchealth.org (K.E.L.); 2Center for Anti-Infective Research and Development, Hartford Hospital, Hartford, CT 06106, USA

**Keywords:** antimicrobial stewardship, transitions of care, outpatient, discharge prescribing

## Abstract

Inappropriate antibiotic use and associated consequences, including pathogen resistance and *Clostridioides difficile* infection, continue to serve as significant threats in the United States, with increasing incidence in the community setting. While much attention has been granted towards antimicrobial stewardship in acute care settings, the transition to the outpatient setting represents a significant yet overlooked area to target optimized antimicrobial utilization. In this article, we highlight notable areas for improved practices and present an interventional approach to stewardship tactics with a framework of disease, drug, dose, and duration. In doing so, we review current evidence regarding stewardship strategies at transitional settings, including diagnostic guidance, technological clinical support, and behavioral and educational approaches for both providers and patients.

## 1. Introduction

Since the initial advent of penicillin in 1928, mankind has been enveloped in a persistent race against the inevitable rise of antibacterial resistance. With antibiotic-resistant infections estimated to affect approximately 2.8 million Americans and causing mortality in 35,000 cases annually, the relevance of this threat has only increased over time [1]. When accounting for the additional collateral impact of *Clostridioides difficile* infection (CDI), associated with antimicrobial use, the toll in the United States rises to over 3 million infections and 48,000 deaths [1]. While most of the attention for resistant pathogens and CDI has traditionally focused on inpatient facilities, recent findings highlight concerning rises in the outpatient community as well.

According to an epidemiologic evaluation of extended-spectrum beta-lactamase (ESBL)-producing *Escherichia coli* (ESBL-EC) and *Klebsiella pneumoniae* (ESBL-KP) within the Southeastern United States between 2009 and 2014, the incidence of ESBL-EC infections increased significantly from 11.1 infections/100,000 patient days in 2009 to nearly double by 2014 at 22.1 infections/100,000 patient days. This increase was largely driven by rises in community-associated and healthcare-associated ESBL-EC infections, with relatively subtle uptrends in hospital-acquired infection [2]. In fact, Enterobacterales as a whole have illustrated marked increases in resistance to commonly prescribed outpatient antibiotics, with reported resistance rates to fluoroquinolones skyrocketing from <1% in the mid-to-late 1990s to upwards of 10–30% in the United States by the mid-2010s depending on geographical location [3]. Similar findings for trimethoprim–sulfamethoxazole resistance in *Escherichia coli* have also been seen, increasing from 17.2% in 2003 to 22.2% in 2012 [4]. International rises in Gram-negative resistance have also been reported, particularly for carbapenem resistance in *Escherichia coli* and *Klebsiella pneumoniae*, which increased significantly in Europe from 0.1% and 8.4% of isolates, respectively, in 2016 to 0.2% and 10% in 2020 [5].

Gram-positive resistance, while not escalating to the same degree as its Gram-negative counterpart, also remains concerning. Despite national targeted activity to decrease the incidence of MRSA infection, longitudinal surveillance data in the United States suggest that the rate of decline has started to slow in more recent years. Between 2005 and 2012, hospital-onset MRSA bloodstream infections demonstrated an annual decline of 17.1%, but then plateaued and did not significantly change from 2013 to 2016. In contrast, community-onset MRSA infections showed an overall less profound reduction, dipping by 6.9% per year from 2005 to 2016 [6]. Comparatively, the percentage of MRSA isolates in Europe was reported to decrease from 2016 to 2020; however, an increasing trend of vancomycin-resistant isolates of *Enterococcus faecium* from 11.6% in 2016 to 16.8% in 2020 was observed [5]. Aligning closely with other pathogens of concern, CDI cases have also demonstrated heightened occurrence in the community setting, increasing from 52.88 infections/100,000 persons in 2012 to 65.93 infections/100,000 persons in 2018, as reported by the Centers for Disease Control and Prevention (CDC) Preventing Emerging Infections Program, whereas healthcare-associated cases actually decreased in the same timeframe from 92.90 to 64.18 infections/100,000 persons [7]. While these findings are not entirely unexpected, they remain concerning given that pathogen resistance and CDI incidence is exacerbated by the routine prescribing and overuse of antimicrobials. 

Nearly 80–90% of human antibiotic use occurs in the outpatient setting, according to public health data gathered from the United Kingdom and Sweden, an estimate that has been presumptively adopted by other countries such as the United States [8]. Within the United States, outpatient prescriptions are estimated to account for approximately 59% of total antibiotic expenditures [9]. Population-based community prescribing data demonstrate an alarming rise in the use of broad-spectrum agents, doubling from 2000 to 2010 in the United States, with older adults serving as the primary recipients of antimicrobial prescriptions [10]. Similar findings have also been reported by the European Centre for Disease Prevention and Control, with the average ratio of community broad-spectrum antibiotic consumption increasing from 2.8 to 3.5 between 2011 and 2020 [11]. In 2020 alone, over 201.9 million outpatient antibiotic prescriptions were written by primary care practitioners (32%), physician assistants and nurse practitioners (31%), and dentists (10%) in the United States, with the most common indications historically reported for ambulatory antibiotics being acute respiratory infections (41%), skin and soft tissue infections (18%), and urinary tract infection (9%) [12,13]. An estimated 28% of outpatient antibiotic prescriptions are considered unnecessary (not indicated at all), with overall total inappropriate use (inclusive of factors such as unnecessary use, inappropriate agent, dosing, and duration) approaching nearly 50%, thus highlighting the significant contribution of the outpatient sector towards the overall burden of antimicrobial use and prompting additional interest for focused stewardship in the community healthcare setting [14].

While most outpatient prescriptions are initiated directly in the community, antibiotic prescribing at transitions of care (TOC) serves as a considerable, though relatively neglected, area when evaluating the totality of outpatient antimicrobial utilization. Approximately half of hospitalized patients are prescribed antibiotics during their inpatient stay, with more than one in eight continued at discharge [15]. Patients treated in the hospital for common infections are likely to complete half of their prescription after leaving the hospital, with many receiving an excess duration prescribed at discharge [16]. With the dynamic shifts in care present at hospital discharge, studies have identified a high prevalence of overall medication errors and care discrepancies occurring at the point of transition, which may contribute to the suboptimal use of antimicrobials. Previous studies evaluating the appropriateness of discharge antimicrobials from acute care facilities reported that an estimated 50–70% of prescriptions were inappropriate in drug choice, dose, or duration [17,18].

Given the continued growing impact of antimicrobial resistance and other ancillary consequences associated with inappropriate antimicrobial use, it is clear that there is a meaningful need for antimicrobial stewardship in the community setting but also at transitions of care. In this review, we summarize current evidence for transitions of care and outpatient stewardship initiatives, outline elements driving suboptimal antimicrobial use, and present interventional strategies for healthcare professionals seeking to strengthen stewardship practices.

## 2. Stewardship Foundations

In 2014, the CDC first introduced the Core Elements of Hospital Antibiotic Stewardship Programs as part of a national call to action in addressing the growing threat of antimicrobial resistance [19]. Acknowledging the need for attention in the outpatient setting, this was quickly followed in 2016 by the release of the CDC Core Elements of Outpatient Antibiotic Stewardship, which served to augment the existing recommendations for other clinical settings and provide a framework for outpatient providers [20]. To date, there is no specific guidance resource for stewardship in transitions of care settings, though synergy of the concepts outlined in the hospital and outpatient core elements (particularly commitment, pharmacy expertise, action for policy and practice, tracking and reporting, and education) can be valuably applied in this novel stewardship area [21]. These core elements advocate for clinicians in this setting to align practice with these concepts by demonstrating accountability towards improving antibiotic utilization, embracing pharmacy involvement, implementing practice guidelines and initiatives in concordance with evidence-based recommendations, tracking and reporting on antibiotic prescribing and utilization data, and providing educational resources to providers, as well as patients and families, on appropriate antibiotic use [20]. While these strategies are universally applicable, stewardship efforts in facilities with limited time and resources may be best optimized by targeting what the CDC refers to as “high-priority conditions”, or situations in which clinicians most frequently deviate from best practices. These encompass conditions for which antibiotics are overprescribed (not indicated), overdiagnosed (diagnosis without fulfilling diagnostic criteria), prematurely prescribed (underuse of watchful waiting or delayed prescribing strategies), misprescribed (wrong agent, dose, or duration selected), or underprescribed (missed diagnoses). Commonly encountered examples of such “high-priority conditions” in transitions of care settings include antibiotic prescribing for acute respiratory infections (overprescribing), treatment of asymptomatic bacteriuria (overdiagnosis), and suboptimal antibiotic regimens, including the selection of a non-preferred agent, failure to account for renal dose adjustments, and excessive duration (misprescribing). In focusing efforts on these high-yield areas, stewards may be able to most effectively maximize the impact of their interventions.

Numerous studies have sought to investigate the impact of various stewardship strategies in order to provide insight for developing practice frameworks and demonstrate support for dedicating resources toward stewardship activities. A 2005 systematic review of 39 studies conducted in primary care and ambulatory clinics examined a variety of activities, including printed educational materials, audit and feedback, educational meetings and outreach visits, financial and system-based changes, patient-based interventions, and multi-faceted interventions, and found that of these actions, a multi-faceted approach involving educational interventions on many levels, targeting healthcare providers, patients, and the general public, had the most magnitude in effect size [22]. These findings were further supplemented by a second systematic review in 2008 of 43 studies conducted in similar outpatient settings, which found that interventions involving active clinician education demonstrated a trend towards improved antibiotic utilization [23]. Other individual studies conducted more recently continue to support the effectiveness of multi-faceted approaches, with one Spanish study working with primary care physicians demonstrating an overall reduction in antibiotic prescribing of 4.2% and a 9.0% decrease in the ratio of broad- versus narrow-spectrum agents attributable to the combined intervention of an hour-long educational outreach paired with an online course and clinical decision support system [24]. Another nonrandomized controlled trial conducted in academic emergency departments in the United States found an improvement in the guideline-adherent treatment of skin and soft tissue infections (SSTI) of 10% and a reduction in antibiotic duration of 26% for sites subject to a multi-faceted intervention consisting of SSTI guideline education with testing, implementation of a clinical treatment algorithm and electronic order set, and clinician audit and feedback [25].

Sparked by the identification of opportunities for improved prescribing in this setting, applications of similar strategies during transitions of care have arisen more recently. In a 2019 systematic review of 40 paired hospital and nursing home sites, Dickinson and colleagues reported that approximately 30% of transitions involved an inappropriate change to the antibiotic plan of care, thus establishing a significant area ripe for improvement [26]. With increasing recognition for its value, preliminary studies assessing stewardship services at transitions of care have illustrated promising results. A pre- and post-intervention study conducted by Zampino and colleagues, evaluating the role of expanded pharmacy stewardship services of prospective audit and feedback on discharge antimicrobials, showed significant improvements in appropriate prescribing, increasing from 47.5% prior to service expansion to 85.2% afterwards. These prescribing improvements were also associated with a reduction in antimicrobial days of therapy from 626.5 to 555 days and a decrease in 30-day hospital readmission rates from 19.7% to 11.5% [27]. By harnessing a focused lens on high-priority conditions and coupling interventions with educational measures, individuals embarking on stewardship in the transitions of care setting have the potential to catalyze meaningful impacts on the optimization of antibiotic use.

## 3. Interventional Approach to Stewardship

Strategies of prospective audit and feedback coupled with education have demonstrated success in improving appropriate antibiotic prescribing and shown benefits towards other ancillary outcomes, such as a reduction in antibiotic days of therapy and hospital readmission rates [20]. In a prospective audit and feedback approach, external stewards review antibiotic therapy for an appropriate indication, drug selection, dose, and duration of therapy and propose recommendations to optimize use, often accompanied by technological and educational evidence-based support. We present a conceptual scaffold to guide transitions of care stewards participating in therapeutic review and education to evaluate the appropriateness of antibiotic prescribing through the lens of four core determinants of therapy summarized in Table 1: disease, drug, dose, and duration.

### 3.1. Disease

At the core of all prescribed therapeutic antibiotic regimens is a targeted disease; thus, the foundation of a suitable antibiotic prescription is first reliant on appropriate diagnosis and triaging a need for antibiotics. Overdiagnosis contributing to inappropriate antibiotic use can occur when a clinical diagnosis is made without fulfilment of diagnostic criteria, or when a diagnosis is made for a disease that would not have caused any symptoms or harm, thus rendering treatment to be superfluous [20,28]. One of the most prevalent examples of overdiagnosis contributing to antibiotic burden is asymptomatic bacteriuria (ASB), which is commonly overdiagnosed as urinary tract infection (UTI). While bacteriuria serves as a shared diagnostic criterion for both ASB and UTI, the presence of additional urinary signs and symptoms is required for a formal diagnosis of UTI. National IDSA guidelines specifically recommend against antibiotic therapy for ASB (with the exception of a few distinct patient populations), as treatment does not improve patient outcomes and instead may be associated with longer durations of hospitalization, adverse drug effects, and the risk of breeding antibiotic resistance [29]. Despite this, a substantial number of patients receive unnecessary antibiotics for ASB, with reports ranging from approximately one third of patients within single-site evaluations and up to 80% across multi-site studies [30,31]. Issues such as the overutilization of altered mental status as a diagnostic symptom (particularly in the elderly or those with underlying cognitive disorders such as dementia), an inability to identify ASB with overreliance on urinary cultures, and a lack of knowledge of indications to treat have been recognized as factors contributing to the high degree of inappropriate antibiotic prescribing and highlight the stewardship opportunities available in tackling the burden of ASB [30]. Thus, a review by the antimicrobial steward to accurately identify the fulfilment of diagnostic criteria with the recognition of indications for treatment is an opportunistic first step in differentiating appropriate vs. inappropriate prescribing. While prospective audit and feedback is a valuable approach in managing individual cases, the approach is time-intensive and may be difficult to conduct for a large population. A more widespread stewardship impact to target systemic practices and provide upstream stewardship support may be accomplished by empowering prescribers to more appropriately diagnose and triage ASB via educational interventions and collaborating with laboratory partners to reduce excessive culture processing. Educational interventions consisting of targeted pocket reference cards, computer station reference postings, and semiannual educational lectures given to primary care providers and nursing staff in a VA long-term care facility demonstrated a sustained reduction in the inappropriate treatment of ASB. The provision of reference cards describing appropriate vs. inappropriate situations in which to send urine cultures and indications for the initiation of antibiotic treatment to nurses and prescribers, respectively, coupled with desktop postings and education sessions, resulted in a reduction in inappropriate urine culture submission from 2.6 to 0.9 per 1000 patient-days, a concomitant decrease in the treatment rate of ASB from 1.7 to 0.6 per 1000 patient-days, and an overall drop in total antimicrobial days of therapy from 167.7 to 117.4 per 1000 patient-days at 6 months after the educational intervention. Notably, these reductions were sustained even at 30 months after beginning education [32]. By educating front-line stakeholders to improve the appropriate recognition and response to suspected ASB, stewards may circumvent inappropriate antibiotic use at the point of diagnostic decision-making and thus lessen the downstream prescribing consequences of overdiagnosis.

Another systemic strategy for ASB stewardship in the diagnostic process includes partnering with the microbiology laboratory to decrease unnecessary screening and the overinterpretation of laboratory results [33]. Diagnostic stewardship interventions such as streamlining criteria for reflex urine cultures from urinalysis and optimizing thresholds for identification and susceptibility testing for urine cultures have demonstrated benefits in cutting superfluous microbiological testing and improving appropriate antibiotic prescribing through judicious laboratory evaluation. After changing the laboratory protocol so that urine reflex culturing was only performed if urinalysis demonstrated > 10 WBC/hpf, a quasi-experimental pre-post study at a VA acute care hospital, emergency department, and two long-term care facilities found a 39% decrease in the rate of urine cultures performed, which likely decreased the number of patients treated for ASB [34]. Similarly, in another study conducted at an acute care hospital by Smith and colleagues, a rise in threshold from > 10^4^ CFU/mL to > 10^5^ CFU/mL for the processing of potential uropathogens was accompanied by lab report commentary stating that organisms present in quantities between 10^4^ and 10^5^ CFU/mL were usually representative of ASB and advising clinicians to call for workup if there was high clinical suspicion for UTI. This intervention led to a decrease in prescribing ASB for patients with colony counts within this range from 38% prior to the intervention to 10% after, with no significant changes in clinical outcomes. Less than 10% of samples were requested for workup on the basis of clinical suspicion, although, when workup was requested, it was found that these patients were more likely to have a UTI (35% vs. 7%), suggesting more thoughtful provider discernment of the clinical indicators of infection [35]. The implementation of organizational activities focused on improving accurate diagnosis via professional education and prudent laboratory procedures is a proactive initiative that may assist in reducing the overall downstream burden of review for antibiotic stewards.

Accurate diagnosis is a critical first step in determining appropriate therapeutic management, with the second being adequate triaging for antibiotic need. Overprescribing occurs when antibiotics are given for conditions in which they are not indicated, such as in the common scenario of acute upper respiratory tract infections. With many acute respiratory tract infections being viral in origin, strategies aiming to dissuade antibiotic initiation may prevent the prescribing of these agents for indications in which they lack a benefit. Even a simple reminder, such as the display of a public commitment poster for judicious antibiotic use in patient examination rooms in a randomized controlled trial across five outpatient primary care clinics, was seen to help influence clinician decision-making and decrease inappropriate antibiotic prescribing for acute upper respiratory tract infections by approximately 20% [36]. Utilization of delayed prescribing practices, in which patients are prescribed antibiotics to take only if symptoms worsen or do not resolve after initial medical consultation, may also be a helpful tool to avoid unnecessary antibiotic use. This strategy has been evaluated for acute upper respiratory tract infections, where studies have demonstrated notable reductions in antibiotic use with delayed prescribing strategies and little difference in symptom control. A multi-site study by Little and colleagues, conducted across 25 primary care practices, divided patients into four delayed prescribing groups (recontact for a prescription, post-dated prescription, prescription collection, or immediate prescription). There was no significant effect on symptom severity or duration between strategy arms but a substantial difference in antibiotic use amongst patients receiving immediate prescriptions (97%) as compared to delayed strategies (ranging from 26% to 39%) [37]. In a similar randomized controlled trial, patients were randomized to one of four prescribing strategies: immediate prescription (given prescription and instructed to start that day), patient-led prescription (given prescription with instructions to take in a few days if no improvement in symptoms), prescription collection (instruction to return in 3 days to collect antibiotic prescription), or no prescription (not offered an antibiotic prescription). While the mean duration of symptoms was shorter for patients who had received prescriptions, antibiotic use was markedly reduced for patients managed with delayed prescribing practices (12.1% in the no prescription group, 23% in the prescription collection group, 32.6% in the patient-led prescription group, and 91.1% in the immediate prescription group), with no differences observed in general health status assessed at 30 days [38]. In the setting of diagnoses in which the clinical benefit of antibiotic use is uncertain, such as in acute upper respiratory tract infections, dissuasion of initial prescribing or delayed prescription strategies may be a useful approach in mitigating unnecessary antibiotic use. While many diagnostic stewardship interventions can be most effectively implemented upstream to minimize the burden of antibiotic prescriptions at discharge, stewards operating at transitions of care are well positioned to confirm the diagnostic indication and assess for the ongoing need of antibiotics at discharge.

### 3.2. Drug

Once an infectious disease warranting treatment is identified, selection of an appropriate drug is the natural next step in building an antibiotic regimen. In the setting of drug selection, stewardship opportunities arise in determining the need to select a drug at all, as well as the nuances of selecting an optimal, as compared to simply suitable, antibiotic agent. While multiple drugs may all adequately treat an indication, stewardship efforts to optimize medication selection based on evidence-directed practice may improve overall appropriate antibiotic use and minimize collateral consequences such as the development of resistance, the risk of which may persist beyond the immediate antibiotic course. One recent case–control study evaluating fluoroquinolone resistance in *Escherichia coli* found that receiving at least one fluroquinolone prescription preceding the diagnosis of resistance was associated with a higher risk of fluoroquinolone-resistant *E. coli* colonization or infection, with the risk of resistance highest in the first year after the antibiotic is taken (OR 2.67) and progressively decreasing to undetectable after two years (OR 1.09) [39]. Thus, judicious drug selection to spare unnecessary or overly broad antibiotic use serves as an important opportunity to potentially curtail some of these repercussions. One promising avenue in which stewards can manifest drug selection reinforcement is through assisting in the building of clinical decision support systems. By integrating guideline recommendations directly within the electronic order process, stewards can help to navigate the initiation of an ideal drug agent. In a retrospective pre- and post-intervention study, a clinical decision support system (CDSS) targeting azithromycin and fluoroquinolone use for various acute respiratory infections incorporated drug-specific guideline recommendations via clickable order entry choices, proposed individualized recommendations based on patient-specific information gathered via electronic medical record mining, and issued documentation of rationale for use. Order entry pathways were built for community-acquired pneumonia, acute bronchitis, acute sinusitis, and non-specific upper respiratory tract infections, in which providers selected diagnostic elements and were led by the CDSS to a drug recommendation if indicated, or, if not, to guidance on how to maintain patient satisfaction when withholding antibiotics at the time of e-prescribing. This intervention significantly reduced unwarranted prescriptions from 22% to 3.3% and improved the proportion of visits with prescribing congruency to guideline recommendations from 0.63 to 0.72 [40]. Another similar approach was utilized in a study conducted within eight primary care clinics, where clinical pathways coupled with patient education materials were implemented for a variety of common infectious conditions, including acute upper respiratory tract infections, acute otitis media, urinary tract infection, skin infections, and pneumonia. Clinical pathways consisting of a decision support algorithm combined with patient education materials were provided to interventional sites in the form of hard binders housed in examination rooms and provider work areas, as well as web access to the documents. The majority of visits were found to be for acute respiratory infections (68.0%–76.4%), which demonstrated a relative reduction in antibiotic treatment of 11.2%, decreasing from 42.7% to 37.9%. Furthermore, the proportion of broad-spectrum antibiotic use for all included indications also demonstrated a relative reduction of 14.4% (26.4% dropping to 22.6%) [41]. Collective data regarding the impact of CDSS on stewardship endeavors across 45 studies in a five-year systematic review found that 90.9% of studies with an endpoint of antibiotic consumption reported a statistically significant decrease in overall consumption, as well as narrowing of the antibiotic spectrum [42]. By investing efforts into crafting and implementing clinical pathways or decision support tools, stewards may not only encourage judicious antibiotic initiation, but also reinforce the thoughtful selection of optimal drug choices to align with guideline-based practice and potentially minimize the risk for adverse collateral effects in the outpatient setting. Interventional opportunities regarding drug selection at discharge further include the facilitation of intravenous to oral conversion, whether sequentially with direct IV to PO conversion of the same drug (i.e., IV to PO metronidazole), with a switch involving IV to PO transition to a different agent in the same drug class (i.e., ceftriaxone to cefpodoxime), or a complete step-down with IV to PO transition between different antibiotic classes (i.e., cefepime to levofloxacin) [43].

### 3.3. Dose

Antimicrobial dosing inaccuracies pose considerable opportunities for improved prescribing practices, with issues arising both from general prescribing errors as well as failure to account for clinical changes occurring during a patient’s course. Previous studies seeking to characterize the incidence of prescribing errors estimate that approximately seven to eight percent of prescriptions contain errors, with antibiotics serving as the leading drug class with prescribing mistakes and reported to account for between 20% and 50% of all observed errors [44,45]. Further analyses of prescribing errors repeatedly highlight incorrect drug dosing and frequencies as the most commonly occurring prescription issues [46,47]. These observations are likely multifactorial and may be attributed to difficulties in navigating the vast range of dosing regimens available for most antibiotics. Unlike many other medications, antibiotics rarely have a standard universal dose. Instead, healthcare professionals must carefully contemplate a complex combination of factors to determine the correct antibiotic dosing, including, but not limited to, indication, pathogen, site of infection, bioavailability, and patient characteristics such as age, gender, comorbidities, and individual renal function. These numerous determinants can pose considerable challenges in selecting an appropriate antibiotic dose, especially in the transitional setting, where the abovementioned factors may have drastically changed between the time of original antibiotic initiation and current prescription. By nature of the management of infectious diseases, patients are often started on antibiotic therapy empirically and must undergo several therapeutic modifications over the course of their treatment based on quickly evolving updates in diagnostic findings, microbiological data, and patient-specific changes such as the ability to tolerate varying routes of administration and fluctuating renal function. Failure to account for such variations may dramatically impact a prescriber’s success in selecting an appropriate antibiotic dose, with the risk of clinical failure and resistance development from underdosing, and an increased danger of adverse drug effects from overdosing. This presents bountiful opportunities for stewards to optimize antibiotic dosing by taking into account the totality of a patient’s course and communicating the most current clinical status and corresponding antibiotic dose to prescribers.

Of the various elements involved in determining antibiotic dosing, evaluation of renal function remains one of the most robust areas in which substantial gaps exist in appropriate prescribing. With many antibiotics (for example, beta-lactams) eliminated via glomerular filtration, dosage adjustment will be required based on either the estimated glomerular filtration rate (eGFR) or creatinine clearance (CrCL). Patients with reduced renal function, whether acutely or chronically, are at the highest risk of antibiotic dosing errors. The majority of investigation into antibiotic dosing with renal impairment has been conducted in the setting of chronic kidney disease (CKD), with studies reporting alarming incidences of inappropriately high doses across various clinical locations. One study conducted by Farag and colleagues found that of 1464 antibiotic prescriptions written for ambulatory care patients with CKD, an average of 64 per 100 prescriptions were dosed in excess [48]. Other studies evaluating inpatient prescribing demonstrated similar findings, with a study by Hu et al. reporting an overall antibiotic dosing error rate of 34% in hospitalized older patients and another by Chahine finding a higher overall rate of 51.6% [49,50]. Despite the clear identification of renal function as a major driver in antibiotic dosing errors, efforts to address this issue by use of automated eGFR reporting have failed to substantially improve prescribing practices. Implementation of eGFR reporting by ambulatory laboratories in Farag’s study did not reduce the rate of antibiotic dosing errors, with a pre-eGFR report error rate of 64 per 100 prescriptions and a post-eGFR report error rate of 68 per 100 prescriptions [48]. Similarly, a different undertaking by Sellier’s group sought to incorporate eGFR dosing alerts into the computerized physician order entry (CPOE) system, where electronic alerts were fired for patients with eGFR <60 mL/min/m^2^ to identify the individual’s latest eGFR, as well as general dose recommendations in the form of adjustment ratios for various eGFR ranges (i.e., eGFR of 15 to 59 mL/min/m^2^, adjusted dosage/ordinary dosage ratio of 50% to 75%). This strategy only resulted in a mild and not statistically significant reduction in the rate of inappropriate prescriptions, with 20.4% of orders being inappropriate in the non-alert period as compared to 18.5% following alert implementation [51]. Taken together, these findings suggest that while dose adjustment per renal function undoubtedly presents a significant area for improvement in antibiotic utilization, simple electronic communication of renal function alone, without a specific dosing recommendation, may be insufficient to curtail improper antibiotic dose selection. In order to overcome this challenge and confer a greater successful impact, these strategies may be improved by elevating their scope through the creation of more detailed order sets capable of providing specific dosing recommendations based on selected indication and patient factors, as well as engaging pharmacists to spearhead these initiatives and communicate recommendations. In a large epidemiologic study by Evans and colleagues, a computer-assisted antibiotic dose monitor was built into the hospital CPOE that automatically reviewed the renal function of patients receiving antibiotics and then sent a report of those possibly receiving excess dosages with alternate dosage recommendations to a pharmacist to review and communicate to prescribers if adjustments were warranted. With the combined efforts of the renal function monitoring system and pharmacist involvement, the percentage of patients receiving excessive doses of antibiotics significantly decreased from 50% pre-intervention to 44% post-intervention. Additionally, the average duration of time for which patients were excessively dosed was significantly reduced from 4.7 days to 2.9 days, resulting in fewer doses (13.4 to 10.9), less cumulative grams of drug (12.0 to 10.4), and lowered drug costs (USD 128 vs. USD 98) as well [52].

While much attention has been given to the occurrence of antibiotic overdosing in the setting of chronic kidney disease, comparatively little has been afforded to the counterpart of acute kidney injury (AKI). By definition, AKI refers to an abrupt deterioration in renal function that may be precipitated by a multitude of factors, including infectious diseases. AKI is a fairly common complication observed in patients presenting for hospitalization, with estimates of approximately 15% occurrence in standard inpatients and reaching up to 60% prevalence amongst those who are critically ill [53]. AKI may progress to acute kidney disease (AKD) if present for > 7 days or even CKD if > 90 days, though, in most instances, renal function will generally recover after the offending etiology has been addressed and supportive measures have been provided. Unlike in CKD, where routine overdosing is the most prevalent issue, AKI presents risks for antibiotic underdosing in the setting of dynamic fluctuations in renal clearance [54,55]. In the most common circumstance where renal function recovers between the time that a patient initially presents and when they are to be discharged, there may exist significant discordance between the originally adjusted dosing scheme and that which is most appropriate for their most current renal status. Underdosing has important implications for not only the risk of treatment failure in the immediate present, but also in the greater scheme of overall pathogen resistance. The impact of antibiotic dosing on resistance development is well described, with several clinical and in vitro studies demonstrating that the dosing strategy can influence the selection for antibiotic-resistant mutants, most commonly in the setting of low antibiotic concentrations selecting for low-level resistance, which may in turn facilitate the emergence of higher degrees of resistance from the surviving bacterial subpopulation [56]. Thus, leveraging technological support and individual performance of patient review to ensure the alignment of antibiotic dosing throughout the clinical course and certainly at the final transitioning stage is imperative in ensuring optimal antibiotic dose selection.

### 3.4. Duration

The final component in crafting an appropriate antibiotic regimen after all other pre-requisite diagnostic, drug choice, and dosing factors are appropriately fulfilled is the determination of the duration of therapy. Duration of therapy serves as an important target for antibiotic stewardship, as unnecessarily prolonged courses increase the individual patient risk for adverse effects and *Clostridioides difficile* infection, in addition to the global risk for the emergence of resistance. Out of all of the four core determinants, the duration of therapy has repeatedly been identified as the leading area in which gaps exist between guideline-recommended therapy and prescribing practice, especially at transitions of care. In one study, discharge antibiotic prescriptions from a 300-bed teaching hospital were examined for appropriateness. Of the 236 prescriptions identified, only 21% were appropriate for the duration prescribed as compared with drug choice (74%), dose (64%), or frequency (64%); additionally, the vast majority (71%) of prescribed durations exceeded guideline-recommended lengths of treatment [57]. These findings mirror those seen in other assessments, such as that by Brower and colleagues, which reported that 81% of patients discharged on antibiotics for urinary tract infections (UTI), community-acquired pneumonia (CAP), or hospital-acquired pneumonia (HAP) received an excessive duration, with an overall median duration of 4 days beyond guideline recommendations [58]. With more than half of discharge antibiotic prescriptions needlessly prolonged, there exist significant opportunities for stewardship efforts to impact the post-discharge durations of therapy at the point of interface between the inpatient and outpatient phases of care.

Various interventions targeting the optimization of antibiotic duration have been evaluated, with strategies employing technological support such as CDSS and pre-populated order sets, as well as personal approaches of classic audit and feedback, being the most studied. Sophisticated CDSS and order sets designed with stewardship goals in mind can serve as highly effective global techniques in improving antibiotic prescribing, particularly with the incorporation of default durations or suggested stop dates. In a multicenter retrospective interventional study, a CDSS consisting of diagnosis-specific antibiotic prescription discharge order sets containing default dose and dispense quantities was implemented within 14 emergency departments. Following implementation of the CDSS, with an overall provider utilization of 60.4%, rates of guideline-concordant antibiotic prescribing increased significantly from 19.1% to 28.1%, with improved appropriateness of antibiotic duration (from 38.8% to 51.1%) serving as the driving factor. These changes were even more pronounced in the post-hoc analysis conducted to assess prescribing practices with versus without the use of the CDSS, which found an improvement in duration from 38.5% to 71.1% [59]. Another CDSS-based intervention by Leo and colleagues designed a multi-faceted CDSS tool comprising preconfigured antibiotic dosing and intervals, a “soft stop” duration of three days with a prompt for re-evaluation and action, and commentary listing guideline-recommended durations of therapy for CAP, HAP, and acute COPD exacerbations. The duration of antibiotic therapy significantly decreased from approximately 9.59 to 7.25 days after adoption of the CDSS intervention, with the guideline-adherent duration improving to 69.3% of patients from a pre-intervention rate of 35.8% [60]. While the development of such supportive technology requires investment upfront, well-designed CDSS and order sets with specific regimen recommendations are advantageous for stewardship programs due to the low need for maintenance activity after creation, ability to capture a greater number of patients compared with individual review, and ancillary potential to educate prescribers and influence practice patterns.

In conjunction with computerized strategies, prospective audit and feedback performed by stewardship teams have also demonstrated success in improving the appropriate antibiotic duration and have the benefit of personal engagement. Direct, verbal recommendations made by infectious diseases pharmacists to primary team prescribers regarding appropriate durations of therapy for 600 patients treated for CAP significantly lowered the median days of therapy from nine in the historical group to six in the post-intervention group, thereby circumventing a total of 586 days of unnecessary antibiotics over a six-month period. This audit and verbal feedback also increased guideline-concordant prescribing rates from 5.6% to 42% [61]. Importantly, audit and feedback techniques do not need to be performed strictly by infectious diseases specialists. Staff pharmacists and other members of the healthcare team can also serve as valuable stewards if provided with the adequate tools and education to recognize suboptimal antibiotic use and intervene. A study by Yogo and colleagues described a methodology in which staff pharmacists were trained by an infectious diseases pharmacist to cross-reference discharge antibiotic prescriptions with institution-specific recommendations developed by the infectious diseases team to align with national guidelines. Prescriptions found to be inconsistent with the recommendations were reviewed with further depth and, when appropriate, prescribing physicians were contacted to suggest adjustments to the discharge prescription. While the total prescribed durations of therapy were reduced only from a median of nine to eight days, this procedure resulted in a statistically significant reduction in discharge days of therapy from a median of six to four days, suggesting that many patients complete the bulk of their appropriate antibiotic course while inpatients, and preventing the unnecessary prolongation of therapy may be the priority focus at discharge [62].

## 4. Education and Feedback

As utilized in many stewardship studies, multifaceted interventions including both a recommendation component coupled with education demonstrate a positive impact on optimizing immediate antibiotic use and also have the added benefit of engaging healthcare personnel directly and promoting overarching practice changes. Recent evaluations regarding prescriber perceptions of their antibiotic knowledge and practice suggest that not only are there opportunities for improved knowledge but also a general desire for more education. In one survey administered to house staff physicians at Johns Hopkins Hospital, 88% of respondents agreed that antibiotics are generally overused, with 72% acknowledging that overuse occurs within their own hospital. Results from the ten-question antimicrobial assessment written by an infectious diseases physician indicated that respondent knowledge of antibiotic use was generally low—the overall average score was 28%, with scorers achieving approximately 37% on the basic questions and 15% on the more advanced. The vast majority of respondents (90%) indicated that they wished for greater antimicrobial education and 22% stated that they had not received any formal antimicrobial education within the past year. These findings indicate that prescribers are generally aware of the issue of suboptimal antibiotic use, appear receptive to educational efforts, and may benefit from increased antimicrobial teaching [63]. There are numerous educational strategies that may effectively influence antibiotic use, including activities such as didactic seminars, interactive online modules, and individual comparative feedback. A prospective controlled trial conducted in four Michigan primary care clinics implemented a half-day educational session for physicians, nurses, and pharmacists focusing on antimicrobial resistance, clinical guidelines, and case studies for bronchitis, pharyngitis, sinusitis, and otitis media. Within the interventional population, overall antibiotic prescribing was found to decrease significantly from 49.9% to 37.6%, including significant declines in prescribing for pharyngitis, otitis media, and non-specific upper respiratory tract infections [64]. Another single-center prospective study aimed at improving prescribing practices for community-acquired pneumonia (CAP) designed a three-tiered interventional approach that included the distribution of a survey to medical staff to assess baseline knowledge and practices related to the management of CAP, an educational lecture presented to medical staff and physicians that included the survey results and further information regarding the appropriate evidence-based duration of therapy for CAP, and a prospective review with direct oral feedback. The median duration of therapy decreased significantly from 10 days to 7 in the post-intervention period, resulting in 148 fewer days of antibiotic therapy overall. Additionally, patients in the post-intervention period were more likely to be discharged without antibiotics (26%) as compared with patients in the pre-intervention period (14%) [65]. In a large systematic review of the literature describing educational programs aimed at improving antibiotic prescribing, analysis of 47 primary care studies and 31 hospital studies found that educational interventions, particularly those which include interactive methods such as educational outreach, workshops, small group discussions, individualized training sessions, practice-based interventions, and case-based learning, were effective strategies to improve guideline adherence, total antibiotic use, prescribing behavior, and the quality of pharmacy practice related to antibiotics [66].

While educational sessions have often traditionally been conducted in a didactic lecture format, utilization of online modules and technological platforms has grown as an attractive option to promote interactive engagement and the application of teaching points. This technique originally surfaced in the earlier phases of medical education targeting student trainees, where web-based distance learning with modules focusing on antimicrobial use and microbial resistance significantly improved assessment performance and student-reported knowledge gain [67]. When applied in the practical clinical setting, incorporation of technology-based training may serve as a viable strategy to target stewardship education towards a larger audience than would be feasible for traditional didactic teaching. In a randomized controlled trial conducted by Butler and colleagues, a blended intervention consisting of several online case scenarios and video modules with interactive exercises focusing on antibiotic resistance and patient case prescribing was implemented across 68 general practice sites, with a goal to evaluate the impact on oral antibiotic prescribing rates. A 4.2% reduction in total oral antibiotic prescribing was seen in the intervention group, with decreased dispensing for all classes of antibiotics, but most profoundly for penicillin V (7.3% reduction) and macrolides (7.7% reduction) [68].

Education-based initiatives can be bolstered further when coupled with performance feedback; by collecting prescribing data following educational outreaches and transparently sharing these results with participants, prescribers may benefit from increased awareness of prescribing practices, evaluate comparative standing when benchmarked amongst peer use, and monitor individual progression over time. Such “behavioral interventions”, as these feedback approaches are commonly referred to, have demonstrated promise in optimizing antibiotic use. In one interventional study, 30 primary care providers (PCPs) attended a regularly scheduled regional meeting in which education on inappropriate antibiotic prescribing consisted of evidence-based guideline review for acute respiratory tract infections and the dissemination of a pocket-guide prescribing algorithm. Following this session, an unblinded email report outlining a peer comparison of inappropriate prescribing rates was emailed to each PCP on a biweekly basis. After one year of this education and feedback strategy, inappropriate antibiotic usage for respiratory tract infections decreased from 51.9% to 31.0%, and further fell to 16.3% following an additional year of continued open feedback [69]. Another study spread across 21 pediatric, family, and internal medicine practices began with education performed at individual clinical staff meetings as well as e-mail communications, and included the sharing of comparative clinic data on antibiotic prescribing for acute respiratory infections to individual site champions, who were then responsible for the academic detailing of outlying providers. This approach resulted in a reduction in overall antibiotic prescribing from a pre-intervention average of 11.5% of visits to 5.8% afterwards [70]. While the majority of behavioral intervention studies have primarily taken place in the outpatient setting and with a focus on acute respiratory infections, there is a substantial opportunity for the similar application of education and performance feedback to be conducted for providers involved with discharge prescribing and for a wider variety of infections.

## 5. Patient Engagement

While prescribers undoubtedly serve as a major target to focus antimicrobial stewardship education, patients also serve as an important population to empower with greater antibiotic understanding. In a world of increasing evidence-based practice and broadening healthcare advancements, historical paternalistic models of medicine have largely been replaced with practices that emphasize shared decision-making and patient-centered care. Not only are patients an important influence in the prescribing process, in many areas of the world where antibiotics are available without prescription, patients may also independently engage in self-medication with antibiotics (SMA), often inappropriately. The prevalence of SMA varies between countries, with ranges reported between 19% and 82% in Middle Eastern countries and 7.3% and 85.59% in Southeast Asian regions [71,72]. Proposed strategies for successfully implementing shared decision-making related to antibiotic consumption include establishing patient expectations from the consultation, negotiating a treatment approach with explanation and reassurance, and the provision of advice regarding symptom duration and relief [73]. These techniques may be valuable tools in coordinating formal antibiotic prescription by healthcare providers as well as dissuading reliance on inappropriate SMA. By engaging patients in education regarding the appropriate prescribing of antibiotics and the associated risks of inappropriate use, both prescriber and patient can mutually establish expectations for care and better optimize antibiotic utilization. In one study conducted by Almarzoky Abuhussain and colleagues, adult patients who presented to the emergency department (ED) across six study sites for the treatment of acute bacterial skin and skin structure infections (ABSSSI) were surveyed upon presentation and 30–40 days after their ED visit regarding preferences for treatment, including factors such as location (hospital, home, outpatient infusion), antibiotic regimen (oral, single IV dose, multiple IV doses), ranking of care priorities (efficacy, administration route, cost, adverse events, treatment location, convenience, doctor’s opinion), satisfaction with care, and interest in single-dose IV antibiotics. Results of the survey found that 40% of patients preferred to be treated at home, and second to “no preference”, the most frequent (39.8%) antibiotic regimen choice was for a single dose of IV antibiotic. Most participants (75.5%) reported that the ED provider did not involve them in any decisions about their care and ultimately 73.4% of patients were admitted to the hospital for treatment, with the majority (75.8%) receiving multiple IV doses of antibiotics [74]. This same group conducted another survey amongst ED providers to elucidate patient factors influencing decisions regarding the treatment of ABSSSI, in which various hypothetical cases were presented and providers were queried regarding their choice in antibiotic regimen and disposition. Findings from the survey revealed that most providers (65.6%) rarely or never asked patients about their antibiotic preference, had variable decisions for hypothetical ABSSSI treatment, and overall coupled hospitalization with the administration of intravenous antibiotics, thus highlighting opportunities for stewardship efforts in addressing antibiotic selection for patients who may be candidates for single-dose and outpatient intravenous antibiotic options [75].

The value of including patients in stewardship efforts has been demonstrated in several published works. In a randomized controlled trial focusing on acute bronchitis in general practice settings, patients provided with an educational leaflet describing the natural course of lower respiratory tract infections and advantages and disadvantages of antibiotic use demonstrated less antibiotic use (47%) as compared with patients who did not receive the leaflet (62%) [76]. Similarly, during a study conducted at an urgent care clinic affiliated with Denver Health Medical Center, a one-hour provider education session focusing on the appropriate utilization of antibiotics for the treatment of adult acute respiratory infections was conducted in conjunction with the display of examination room posters and a patient-directed interactive computerized education (ICE) module. Depending on whether they participated in the ICE module, patients were classified as having limited (no ICE exposure) or full (completed ICE) intervention exposure. Following this dual-pronged intervention, the proportion of patients receiving antibiotics for acute bronchitis was seen to decrease from 58% to 30% in the limited intervention group and even further to 24% in the full intervention group. For nonspecific upper respiratory tract infections, antibiotic prescriptions decreased from 14% to 3% and 1% in the limited and full intervention groups, respectively [77]. As most currently available studies evaluating patient education take place in the outpatient community, with a primary focus on self-limiting respiratory conditions, there exist considerable openings for expansion into the transitions of care setting and consideration for other infectious syndromes. From a transitions of care perspective, opportunities for greater patient engagement include counseling patients on the risks versus benefits of antibiotic therapy and encouraging collaborative discussion regarding the need for antibiotics at the point of discharge.

## 6. Future Frontiers

Overall, there exists an abundance of possibilities for stewardship expansion into the transitions of care setting, with a variety of opportunities across all four facets of antibiotic determinants of therapy. However, in addition to identifying such areas of opportunity and interventional measures to address them, the steward must also be armed with an understanding of the factors that may initially pose barriers in the pursuit of their initiatives. A systematic review of 35 studies seeking to understand intrinsic and extrinsic factors driving physician antibiotic behavior found that complacency in prescribing antibiotics to fulfill perceived patient expectations and fear of complications or patient loss were the most dominant physician attitudes reported, and sociodemographic factors such as university education also influenced clinical practice [78]. As discussed in depth previously, clinician knowledge gaps serve as an important driver of suboptimal antimicrobial use and may be mitigated by identifying specific areas of deficiency for which to conduct educational endeavors and perform evaluative behavioral interventions. Additional challenges seen in implementing stewardship strategies, particularly in the transitions of care setting, include the perception of patient expectations and uncertainty of patient clinical oversight. Pressure by patients, real or otherwise perceived, is a common concern raised by providers writing for antibiotic prescriptions. In a qualitative study of primary care clinicians performed by Jeffs and colleagues, prescribers universally reported feeling pressured to prescribe antibiotics by patients and felt that those who were not prescribed antibiotics expressed greater dissatisfaction with their care and were more likely to visit another clinic [79]. This pressure is a powerful influencer in the prescribing process; another survey of general practitioners found that approximately 55% felt pressure to prescribe an antibiotic even if they were unsure if a prescription was necessary, with 44% admitting to prescribing antibiotics to facilitate ending the patient visit and 45% prescribing antibiotics for viral infections knowing that they would not be effective [80]. In light of this considerable challenge, studies have sought to investigate patient expectations regarding antibiotic use. One small study consisting of 31 patient interviews found that patients described antibiotics as trusted agents with widespread use, and approximately half expressed expectations to receive an antibiotic, particularly patients presenting with recurrent UTIs [81]. Another prospective observational cohort study performed by Ong and colleagues across ten academic EDs interviewed patients who presented with a chief complaint consistent with a respiratory infection. Forty-eight percent of patients expected an antibiotic prescription during their visit, with 30% reporting treatment with antibiotics in the past for a similar illness. These patients were statistically more likely to expect antibiotics and believe that antibiotics were necessary for their current illness as compared with patients who had not received antibiotics previously for similar conditions. Additionally, 89% of patients knew that antibiotics were effective against bacteria, but 69% also believed them to be effective against viruses [82]. Other studies have reported varying rates of patient antibiotic expectations ranging from 16% to 22% depending on clinical complaint [83,84] but similarly highlight historical expectations of care and gaps in understanding antibiotic utility as core themes driving patient pressure for antibiotics. Another considerable barrier to stewardship interventions at transitions of care includes the perception of patient vulnerability at the point of discharge, at which point patients may no longer be as closely monitored and the effects of clinical decisions may not be as easily ascertained. In this situation, clinicians may err on the side of prescribing broader or longer antibiotic regimens out of an overabundance of caution and effort to prevent readmission, without weighing the concomitant short-term and long-term risks of inappropriate antibiotic use [85]. Many strategies discussed previously in this paper can be utilized to temper the impact of patient pressure and perceived vulnerability on inappropriate antibiotic prescribing at transitions of care, including prospective audit and feedback evaluating the need for antibiotic use based on diagnosis, reinforcement of drug selection, dosing, and duration of therapy in clinical decision support tools, and, importantly, strengthening both provider and patient education regarding the role of antibiotics in their medical care.

Increasingly, antibiotic prescribing at transitions of care is growing as an important sector in the ongoing battle against antimicrobial resistance. For the emerging steward, the established literature and CDC guidance support targeting efforts on the most high-yield disease states for which antibiotics may be inappropriately prescribed, including acute respiratory tract infections, urinary tract infections, and skin and soft tissue infections. Across the board, strategies involving the integration of clinical decision support tools with the implementation of audit and feedback coupled with peer comparison and accountability can help stewards in the transitions of care setting to optimize antibiotic prescribing with attention to disease, drug, dose, and duration. Further progression of stewardship activities in this realm should ideally emphasize patient engagement and education, as the patient remains the singular constant across the continuity of care and may serve as a powerful advocate in their medical management and an influencer of antibiotic use. Additionally, collaboration with other stakeholders, such as electronic health record vendors, insurers, and local health departments, may enhance stewardship efforts through improved monitoring and the potential to more accurately measure antibiotic use in this dynamic setting. Finally, stewards working in the transitions of care setting are uniquely poised to partner with both inpatient and outpatient care teams, where increased communication between these settings of care may reduce the potentiation of errors, provide a more comprehensive view of a patient’s care, and optimize overall medical management.

## Figures and Tables

**Table 1 antibiotics-11-01027-t001:** Interventional stewardship strategies.

Therapeutic Consideration	Specific Targets	Example Interventions	Intervention Role at TOC
Disease	Asymptomatic bacteriuria Acute upper respiratory tract infections	Diagnostic stewardship via collaboration with microbiology lab workflow Support for delayed prescribing practices Prospective audit and feedback at point of discharge to confirm clinical criteria	Reduce downstream burden of antibiotic use at discharge Confirm persistence of indication at point of transition
Drug	Community-acquired pneumonia Urinary tract infections	Integration of order pathways and clinical decision support tools into electronic health record	Optimize selection of most targeted agent with minimal collateral damage
Dose	Chronic kidney disease Resolution of acute kidney injury	Incorporation of renal dosing recommendations within computerized provider order entry Dedicated renal function reporting with standard review	Re-evaluate alignment of current dosing regimen with most up-to-date renal function and clinical status
Duration	Community-acquired pneumonia (typically 5 days in duration) Urinary tract infection (typically 5–10 days in duration) Skin and soft tissue infection (typically 5–7 days in duration)	Build duration recommendations in existing order sets Antibiotic time-out clinical decision support service	Assess duration of therapy completed inpatient and appropriate remaining days of therapy for discharge

## Data Availability

Not applicable.

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
