# Peer review of "Antimicrobial Stewardship at Transitions of Care to Outpatient Settings: Synopsis and Strategies"

_antibiotics, 2022, doi:10.3390/antibiotics11081027_

Round 1

Reviewer 1 Report

Dear authors,

It was a pleasure to review this article, it is very well structured and written, the topic is interesting and important.

My minor suggestions for improving the text are:

Introduction

I would suggest that you add data on consumption and resistance to antibiotics outside the USA, for example data from WHO and ECDC.

Patient engagement

I find it important to mention there is data describing self-medication with antibiotics (SMA) in many regions in the world (for example Alhomoud et al International Journal of Infectious Diseases 2017., Jirjees F Antibiotics. 2022.,  Nepal G, Cureus. 2018) and that patients should generally be educated on the danger of this practice.

In the introduction part from my point of view there is a need to better describe the defined TOC (transitions of care) process and standards that exist. In that context the phrase TOC Intervention Role in the Table 1 may be inappropriate (please consider for example:   Intervention role at TOC)

Best regards

Author Response

  • I would suggest that you add data on consumption and resistance to antibiotics outside the USA, for example data from WHO and ECDC.
    • Thank you for your suggestion – given the international readership of Antibiotics, we have added information regarding international trends in resistance [lines 42-45, lines 53-56] as well as reported rates of antibiotic consumption [lines 73-75].
  • I find it important to mention there is data describing self-medication with antibiotics (SMA) in many regions of the world (for example Alhomoud et al International Journal of Infectious Diseases 2017., Jirjees F. Antibiotics. 2022., Nepal G, Cureus. 2018) and that patients should generally be educated on the danger of this practice.
    • We appreciate the highlighting of this issue and have now included commentary regarding the incidence of SMA in various international regions utilizing some of the literature suggested above as well as importance of patient engagement in dissuading inappropriate SMA [lines 623-632].
  • In the introduction part from my point of view there is a need to better describe the defined TOC (transitions of care) process and standards that exist. In that context the phrase TOC Intervention Role in the Table 1 may be inappropriate (please consider for example: Intervention role at TOC)
    • Thank you for this feedback. We are unfortunately unable to describe TOC standards for the purposes of this article, as these various practices vary tremendously not only between individual institutions but also internationally. We have however, re-titled Table 1 as suggested.

Reviewer 2 Report

Thank you for asking me to review the manuscript “Antimicrobial stewardship at transitions of care to outpatient settings: synopsis and strategies” by Elaine Liu et al.

This interesting and relevant narrative, non-systematic, review deals with a hugely important and understudied topic, namely antibiotic stewardship in the interface between in- and outpatient care.

Unless the non-systematic and somewhat subjective design of the study, after some revisions (some references are inaccurate and I would suggest to develop some more reasonings, as highlighted below), I would recommend it for publication in Antibiotics, as the authors develop their reasoning with competence and a red line can be drawn throughout the manuscript: first, authors introduce the topic thoroughly; second, they highlight the need of dealing with AB stewardship at the transition of care (TOC), i.e. the majority of patients are discharged from the hospital with an AB prescription and, the majority of those are discharged with an inappropriate prescription(!); third, authors discuss the foundations of stewardship and the specific targets of stewardship interventions at the TOC; fourth, they conclude their article with appropriateness, leaving the door open for further research.

General points.

I agree with you that at the transition of care both, hospital and primary care setting have to be considered. I strongly suggest you to expand the primary care literature you are dealing with, you seemed to have missed many important studies in primary care, e.g.: very recent https://www.clinicalmicrobiologyandinfection.com/article/S1198-743X(22)00330-5/fulltext when dealing with UTI, 

Dealing with CDSS, consider a recent review you may have overlooked: https://pubmed.ncbi.nlm.nih.gov/31342180/

You can find more relevant suggestions below.

Point-by point review to the authors.

- In the introduction (line 23, line 26, line 39, line 101 , etc.) you refer to the US, the CDC etc. I understand that you are much interested on what is happening/recommended in the US but for an international readership, I suggest you to open the perspective to the rest of the world, at least to Europe (much efforts have been done in Europe in antibiotic stewardship that produced relevant studies). This would be 1) more interesting for your readers; 2) much more coherent with your review, as you often cite non-US authors (e.g. Paul Little, Chris Butler, etc.). 

- Line 57: not only “inappropriate”, also appropriate prescriptions lead to AMR, please change accordingly.

- Lines 58-59 “With the community setting…” Sentence makes little sense here in the flow, consider removing

- Line 61: ref 7 is not appropriate as the CDC reference is referring to studies of PHE and performed in Sweden, please use primary evidence 

- Line 68 ref 10 is not appropriate, it is not referring to 2005

- Line 66: “An estimated 28% of these…”, this is not true, ref. 11 is not referring to prescriptions you mention in the sentence before (which is referring to other prescriptions). Please revise.

- Lines 78- 80: not clear what you mean, ref. 13 is dealing with the fact that approx. 93% of patients received to many days of treatment

- lines 101 and followings (sentence 1 and 2 of the paragraph) need references

- references 18 and 19 are quite old studies. I suggest you to consider to expand this section with new studies such as https://www.ncbi.nlm.nih.gov/pmc/articles/PMC7722452/, https://pubmed.ncbi.nlm.nih.gov/33139143/,https://pubmed.ncbi.nlm.nih.gov/30214718/

- Lines 161-164: sentence needs reference(s)

- Lines 175 and following: this is a wrong definition of overdiagnosis. Overdiagnosis is not about not fulfilling diagnostic criteria, this is the definition of a wrong diagnosis, overdiagnosis happens when you diagnose a disease correctly but there was no need to diagnose it, i.e. the disease probably would have never caused any symptoms to the patient and therefore treatment has no benefits or could cause harms. Please rephrase ore delete.

- Line 181: IDSA guidelines cited without any reference. Consider citing also international guidelines.

- Lines 192-194: no references. Are these your opinions? Please add references, this is not an opinion paper. If reference 24 is the one you are talking about, this is “far away”, consider shortening the text.

- Lines 252 and followings: URTI are not a setting, primary care is the setting, URTI is the disease the doctors in that setting were dealing with

-  Line 322 sentence needs a reference

- when dealing with “dose” (par. 3.3.) it is unclear to me why you are discussing only GFR of AKI. That the dose needs to be adapted to the patient is true (maybe somewhat obvious), please consider that also age, gender and other comorbidities are relevant, this should be mentioned and shortly discussed. Most importantly, the dose of the antibiotic is closely related to AMR, please consider discussing https://pubmed.ncbi.nlm.nih.gov/34726708/. The concept of dosage related selection is also relevant for this paragraph https://academic.oup.com/cid/article/45/Supplement_2/S129/285292. I would suggest you to discuss recent relevant evidence dealing with the decay of E.coli resistance, a burdensome pathogen, which is highly dependent of the dose of the antibiotic administered before (up to several years, depending on the substance and the resistance): https://pubmed.ncbi.nlm.nih.gov/35740228/ & https://pubmed.ncbi.nlm.nih.gov/35740228/

- Ref. 47 needs review, it is not retrievable at all

- Lines 486-491: not clear, I can not understand what is the meaning.

- Lines 532 and followings, consider the following hugely relevant studies which deals with educational interventions, including online interventions, in primary care: https://pubmed.ncbi.nlm.nih.gov/25511932/,https://pubmed.ncbi.nlm.nih.gov/31342180/

- When dealing with patient engagement (5.): https://www.clinicalmicrobiologyandinfection.com/article/S1198-743X(15)30084-7/fulltext

- In the paragraph “future frontiers” (6.), whe dealing with patient expectations and prescribing, consider discussing https://www.ncbi.nlm.nih.gov/books/NBK570462/, and https://pubmed.ncbi.nlm.nih.gov/23127482/

 - Line 693 and following: I very much agree with these conclusions, this is hugely important.

I would be happy to review the revised manuscript.

Best wishes

Author Response

  • I agree with you that at the transition of care both, hospital and primary care setting have to be considered. I strongly suggest you to expand the primary care literature you are dealing with, you seemed to have missed many important studies in primary care, e.g.: very recent https://www.clinicalmicrobiologyandinfection.com/article/S1198-743X(22)00330-5/fulltext when dealing with UTI.
    • Thank you for your valuable feedback. We absolutely agree that both hospital and primary care settings are vital areas when dealing with transitions of care. That said, the intent of this review is to specifically focus on the point of transition from an inpatient to outpatient setting and opportunities for stewardship during this discrete time, as this is a setting that has not been readily addressed by existing stewardship guidance. A discussion of primary care is certainly important but is a substantial topic which the authors feel is deserving of its own focus and is outside the intended scope of this piece for an in-depth review.
  • Dealing with CDSS, consider a recent review you may have overlooked: https://pubmed.ncbi.nlm.nih.gov/31342180/. You can find more relevant suggestions below.
    • Thank you for providing this additional literature – we have incorporated these findings in the CDSS section of the manuscript [lines 340-343].
  • In the introduction (line 23, line 26, line 39, line 101 , etc.) you refer to the US, the CDC etc. I understand that you are much interested on what is happening/recommended in the US but for an international readership, I suggest you to open the perspective to the rest of the world, at least to Europe (much efforts have been done in Europe in antibiotic stewardship that produced relevant studies). This would be 1) more interesting for your readers; 2) much more coherent with your review, as you often cite non-US authors (e.g. Paul Little, Chris Butler, etc.).
    • We appreciate the consideration for our international colleagues and have added more literature and commentary regarding global data throughout the introduction [lines 42-45, lines 53-55, lines 73-75].
  • Line 57: not only “inappropriate”, also appropriate prescriptions lead to AMR, please change accordingly.
    • Thank you, we have adjusted the verbiage to read “… exacerbated by routine prescribing and overuse of antimicrobials” [line 63].
  • Lines 58-59 “With the community setting…” Sentence makes little sense here in the flow, consider removing.
    • We appreciate the feedback and have removed this sentence.
  • Line 61: ref 7 is not appropriate as the CDC reference is referring to studies of PHE and performed in Sweden, please use primary evidence.
    • Thank you for your comment. We have re-phrased the sentence to clarify that the original outpatient antibiotic use data was gathered from the UK and Sweden with subsequent adoption by the US (as the US does not appear to have its own specific data via primary evidence available) [lines 66-68]. We have adjusted the US expenditure data based on more recent primary literature [lines 68-70].
  • Line 68 ref 10 is not appropriate, it is not referring to 2005.
    • Thank you for your feedback. We have adjusted the verbiage to clarify that the listed indications are those historically reported.
  • Line 66: “An estimated 28% of these…”, this is not true, ref. 11 is not referring to prescriptions you mention in the sentence before (which is referring to other prescriptions). Please revise.
    • Thank you, we have adjusted the sentence to more accurately state “An estimated 28% of outpatient antibiotic prescriptions…” [line 80].
  • Lines 78- 80: not clear what you mean, ref. 13 is dealing with the fact that approx. 93% of patients received to many days of treatment.
    • Thank you for your comment. The intended takeaway from ref 13 (now ref 16) is that not only are many patients receiving too many days of therapy (DOT) overall (67.8%), majority of these “excess” days were prescribed at discharge (93%) and completed outpatient. This suggests that many patients who are initially hospitalized for these common infections (CAP, UTI, SSTI) tend to start their therapy in the inpatient setting, but then are often extraneously continued at discharge (potentially due to failure to accurately account for inpatient DOT), which highlights TOC as an important point to evaluate appropriate durations in therapy. We have adjusted the line to clarify that many patients receive “… an excess duration prescribed at discharge” [line 93].
  • Lines 101 and followings (sentence 1 and 2 of the paragraph) need references.
    • Thank you, references have been added (ref 19 and ref 20).
  • References 18 and 19 are quite old studies. I suggest you to consider to expand this section with new studies such as https://www.ncbi.nlm.nih.gov/pmc/articles/PMC7722452/, https://pubmed.ncbi.nlm.nih.gov/33139143/,https://pubmed.ncbi.nlm.nih.gov/30214718/.
    • Thank you for your valuable addition. We have incorporated some of this literature regarding impact of multi-faceted stewardship initiatives (ref 24 and ref 25).
  • Lines 161-164: sentence needs reference(s).
    • Thank you, ref 28 has been added [lines 178-181].
  • Lines 175 and following: this is a wrong definition of overdiagnosis. Overdiagnosis is not about not fulfilling diagnostic criteria, this is the definition of a wrong diagnosis, overdiagnosis happens when you diagnose a disease correctly but there was no need to diagnose it, i.e. the disease probably would have never caused any symptoms to the patient and therefore treatment has no benefits or could cause harms. Please rephrase ore delete.
    • We appreciate this commentary – the definition of overdiagnosis originally used in the paper is in direct alignment with the definition provided explicitly by the CDC’s Core Elements of Outpatient Antibiotic Stewardship and was utilized in this context. However, acknowledging the validity of the reviewer’s clarification, additional verbiage has been added to further define overdiagnosis in the traditional manner [lines 194-195].
  • Line 181: IDSA guidelines cited without any reference. Consider citing also international guidelines.
    • Thank you, we have added the IDSA reference (ref 30) and in reviewing other international guidance, found that other global associations (ie. ESCMID) also refer to these 2019 IDSA guidelines within their own organization resources.
  • Lines 192-194: no references. Are these your opinions? Please add references, this is not an opinion paper. If reference 24 is the one you are talking about, this is “far away”, consider shortening the text.
    • Thank you for your feedback. While lines 192-194 (now 211-213) are not necessarily statements directly gleaned from any particular reference, the intent of these comments was to present a logical opportunity for diagnostic stewardship initiatives and segue from individual diagnostic review to more widespread interventions. Thus, we have not added a reference but instead have respectfully adjusted the verbiage to read “…recognition of indications for treatment is an opportunistic first step…” [lines 212-213] to emphasize this.
  • Lines 252 and followings: URTI are not a setting, primary care is the setting, URTI is the disease the doctors in that setting were dealing with.
    • Thank you, the text has been adjusted to read “This strategy has been evaluated for acute URTI…” [line 272].
  • Line 322 sentence needs a reference.
    • Thank you, reference 44 has been added [lines 347-352].
  • When dealing with “dose” (par. 3.3.) it is unclear to me why you are discussing only GFR of AKI. That the dose needs to be adapted to the patient is true (maybe somewhat obvious), please consider that also age, gender and other comorbidities are relevant, this should be mentioned and shortly discussed. Most importantly, the dose of the antibiotic is closely related to AMR, please consider discussing https://pubmed.ncbi.nlm.nih.gov/34726708/. The concept of dosage related selection is also relevant for this paragraph https://academic.oup.com/cid/article/45/Supplement_2/S129/285292. I would suggest you to discuss recent relevant evidence dealing with the decay of E.coli resistance, a burdensome pathogen, which is highly dependent of the dose of the antibiotic administered before (up to several years, depending on the substance and the resistance): https://pubmed.ncbi.nlm.nih.gov/35740228/
    • Thank you for this valuable feedback. We have added age, gender, and comorbidities as factors to consider related to dosing (lines 368-369), though the overall item of discussion still focuses primarily on renal function as this value is quantifiable and is associated with clearly delineated dose adjustments for many antibiotics, whereas the other patient factors listed as above are certainly relevant but are more abstract clinical considerations. We have additionally incorporated greater discussion regarding dose-related resistance selection (lines 444-450) utilizing some of the above suggested literature. We have respectfully elected to not incorporate the first suggested reference (2021 CAP-IT trial), as the findings of this study primarily deal with investigation of low-dose versus high-dose amoxicillin and duration of therapy for pediatric CAP, with secondary outcomes of pneumoniae colonization resistance demonstrating no significant difference based on dose or duration. While this article is nonetheless interesting in the greater scheme of delineating ideal dose/duration for pediatric CAP in the clinical setting, the authors feel its conclusions do not align strongly within this paper focusing on stewardship strategies. The second suggested reference has been gratefully included (lines 444-450, ref 57). The third suggested reference involving E. coli fluoroquinolone resistance and decay has also been included (lines 306-313, ref 40), though re-located to the drug selection portion of the manuscript as the authors feel that the scope of this paper lends itself more robustly in the discussion of stewardship related to drug selection, as individual dosing schemes were not addressed and the conclusion regarding resistance development was found to be related to cumulative drug exposure.
  • 47 needs review, it is not retrievable at all.
    • Thank you for highlighting this oversight, the doi has been corrected accordingly (ref 60).
  • Lines 486-491: not clear, I can not understand what is the meaning.
    • Thank you, we have removed some excess verbiage to facilitate clarity; the ultimate takeaway from the referenced study is that many patients discharged from the hospital have already completed majority of their antibiotic course while inpatient, thus the stewardship opportunity at discharge is to prevent unnecessary (potentially inadvertent) prolongation of their treatment course. This is demonstrated by the study findings of largely similar total durations of therapy (9 vs 8 days), but a more profound reduction in discharge days of therapy (6 vs 4 days) with intervention (lines 519-524).
  • Lines 532 and followings, consider the following hugely relevant studies which deals with educational interventions, including online interventions, in primary care: https://pubmed.ncbi.nlm.nih.gov/25511932/,https://pubmed.ncbi.nlm.nih.gov/31342180/
    • Thank you for the suggestions, we have added both pieces of literature into the manuscript. The first article describing a systematic review of educational interventions has been incorporated in lines 567-573 (ref 67), while the second piece is the same systematic CDSS review described previously in lines 340-343 (ref 43).
  • When dealing with patient engagement (5.): https://www.clinicalmicrobiologyandinfection.com/article/S1198-743X(15)30084-7/fulltext.
    • Thank you, findings from this helpful piece have been added in the discussion regarding patient shared-decision making [lines 627-631, ref 74].
  • In the paragraph “future frontiers” (6.), when dealing with patient expectations and prescribing, consider discussing https://www.ncbi.nlm.nih.gov/books/NBK570462/, and https://pubmed.ncbi.nlm.nih.gov/23127482/.
    • Thank you, greater discussion including these articles have been added to the text in lines 688-692 (ref 79) and lines 709-712 (ref 82).
  • Line 693 and following: I very much agree with these conclusions, this is hugely important.
    • Thank you for your comment, we appreciate the support and concur that individuals in the TOC setting have untapped potential to facilitate patient care across varied healthcare settings and improve continuity and clarification of therapy.

Round 2

Reviewer 2 Report

Dear authors and dear editor,

I very much appreciated reading the revised version of this relevant review. The authors did a great work in the revision of an already good and interesting paper.

I encourage the editor to publish this expert contribution in its current version and I wish the authors good luck in their effort to develop this interesting frontier of antibiotic stewardship.

Kind regards